# *Cis*-interactions between Notch and its ligands block ligand-independent Notch activity

**William Hunt Palmer, Dongyu Jia, Wu-Min Deng\***

Department of Biological Science, Florida State University, Tallahassee, United States

**Abstract** The Notch pathway is integrated into numerous developmental processes and therefore is fine-tuned on many levels, including receptor production, endocytosis, and degradation. Notch is further characterized by a twofold relationship with its Delta-Serrate (DSL) ligands, as ligands from opposing cells (*trans*-ligands) activate Notch, whereas ligands expressed in the same cell (*cis*-ligands) inhibit signaling. We show that cells without both *cis*- and *trans*-ligands can mediate Notch-dependent developmental events during *Drosophila* oogenesis, indicating ligand-independent Notch activity occurs when the receptor is free of *cis*- and *trans*-ligands. Furthermore, *cis*-ligands can reduce Notch activity in endogenous and genetically induced situations of elevated *trans*-ligand-independent Notch signaling. We conclude that *cis*-expressed ligands exert their repressive effect on Notch signaling in cases of *trans*-ligand-independent activation, and propose a new function of *cis*-inhibition which buffers cells against accidental Notch activity.

## Main text

Canonical Notch signaling begins when the Notch receptor receives a stimulus from a DSL-type ligand (Delta [Dl] or Serrate [Ser] in Drosophila) in an adjacent cell, which leads to γ-secretase-dependent cleavage of Notch, and translocation of the intracellular domain—$N^{ICD}$— into the nucleus to act as a transcriptional co-activator (***de Celis, 2013***). Notch may also be activated in a non-canonical, DSL-ligand independent manner (***Hori et al., 2012***). DSL ligands can *cis*-inhibit ligand-dependent Notch activation when expressed in the same cell as the receptor (***Micchelli et al., 1997***; ***Del Álamo et al., 2011***). However, the possibility of a relationship between DSL-ligand independent Notch activation and *cis*-expressed ligands has not been explored.

The developing Drosophila egg chamber is a convenient model for dissecting the effects of Notch ligands in *cis* and in *trans*, as Dl is the sole signaling source and the signal sending and receiving cells can be easily distinguished (***Deng et al., 2001***; ***López-Schier and St Johnston, 2001***). (***Figure 1—figure supplement 1*** provides a brief schematic depiction of the stages of early oogenesis.) At oogenesis stage 7, Notch signaling is activated in the somatic follicle cells by a robust germline Dl upregulation, which leads to the expression of *Hindsight* (*Hnt*), downregulation of *Cut*, and the polyploidization of the follicle cells (***Deng et al., 2001***; ***López-Schier and St Johnston, 2001***; ***Sun and Deng, 2005***, ***2007***) (***Figure 1A***). When *Dl* germline mutant clones were generated (i.e., *trans*-activation was removed), the follicle cells failed to downregulate *Cut* expression, which persisted past stage 7, indicative of a failure to activate Notch (***Figure 1B***). In contrast, *Dl* follicle cell mutant clones show precocious *Cut* downregulation at stage 6 attributable to the relief of *cis*-inhibition (***Poulton et al., 2011***) (***Figure 1C***). Surprisingly, *Dl* mutant clones in the follicle cells bordering *Dl* mutant clones in the germline (i.e., a germline with no signaling source, herein referred to as *Dl-/Dl-* cells) show correct *Hnt* and *Cut* expression from stage 7 (***Figure 1D,E***, ***Figure 1—figure supplement 2A,B***). These *Dl-/Dl-* clones also correctly transit into the endocycle, as their nuclear volumes are similar to wild-type follicle cells in the later stages of oogenesis after polyploidization (***Figure 1F,G***), whereas cells neighboring

**\*For correspondence:** wumin@bio.fsu.edu

**Reviewing editor**: Utpal Banerjee, University of California, Los Angeles, United States

**eLife digest** Many biological processes require cells to send messages to one another. Typically, this is achieved when molecules are released from one cell and make contact with companion molecules on another cell. This triggers a chemical or biological reaction in the receiving cell.

One of the most common examples of this is the Notch pathway, which is used throughout the animal kingdom and plays an important role in helping cells and embryos to develop. The Notch protein itself is a 'receptor' protein that is embedded in the surface of a cell, and relays signals from outside the cell to activate certain genes inside the cell. In fruit flies, two proteins called Serrate and Delta act as 'ligands' for Notch—by binding to Notch, they can change how this receptor works.

If Serrate or Delta are present on the outside of one cell, they can activate Notch (and hence the Notch signaling pathway) in an adjacent cell. However, if the Serrate or Delta ligands are present on the surface of the same cell as Notch they turn the receptor off, rather than activate it. Notch can also work without being activated by Serrate or Delta, but whether the ligands can inhibit this 'ligand-independent' Notch activation if they are on the surface of the same cell as the Notch receptor was unknown.

Palmer et al. study Notch signaling in the fruit fly equivalent of the ovary, in cells that are naturally deficient in Serrate and from which Delta was artificially removed. The Notch protein was activated when these ligands were not present. Furthermore, the developmental processes that are activated by Notch were able to proceed as normal when triggered by ligand-independent Notch signaling. In total, Palmer et al. investigated three different types of fruit fly cell, and found that ligand-independent Notch signaling can occur in all of them.

Reintroducing Delta to the same cell as Notch turns the receptor off, suggesting that ligands on the surface of the same cell as the receptor can inhibit ligand-independent Notch activity. Many genetic diseases and cancers have been linked to Notch being activated when it should not be; therefore, understanding how Notch is controlled could help guide the development of new treatments for these conditions.

*Dl-/Dl-* follicle cell clones (retaining a *cis*-ligand but without a *trans*-ligand) are comparable to wild-type cells before entry to endocycle (**Figure 1F,G**). Removal of both *cis*- and *trans*-Dl through knockdown of Dl by RNA interference (RNAi) simultaneously in the germline and soma confirmed this finding (**Figure 2—figure supplement 1A,B**). Together, these observations provide evidence that follicle cells without both *cis*- and *trans*-ligand sources can still enter the endocycle stages of oogenesis. This back-up route to the endocycle is not a co-option of Ser in place of Dl, as *Dl^RevF10^Ser^Rx82^* double clones recapitulated the *Dl-/Dl-* phenotype (**Figure 1E**, **Figure 1—figure supplement 2A**).

To determine whether the entry into the endocycle in *Dl-/Dl-* follicle cells still requires the function of Notch, we implemented the mosaic analysis with a repressible cell marker (MARCM) system (**Lee and Luo, 2001**). The MARCM system enables us to create mutant clones while driving expression of a UAS transgene specifically in those clonal cells. *Dl-/Dl-* clones driving expression of *Notch^RNAi^* show a significantly higher proportion (p < 0.0001) of late *Cut*-expressing cells than the *Dl-/Dl-* clones alone, indicating that Notch is still required for the mitotic-to-endocycle switch (**Figures 1D and 2A**, **Figure 1—figure supplement 2C**, **Figure 2—figure supplement 1C**, **Supplementary file 1**). Likewise, MARCM clones for the null allele of Suppressor of Hairless (Drosophila Notch transcriptional effector), *Su(H)^47^*, in RNAi-induced *Dl-/Dl-* clones also show late *Cut* expression (p < 0.0001) (**Figure 2B**, **Figure 1—figure supplement 2C**, **Supplementary file 1**) in comparison with RNAi-induced *Dl/Dl-* clone controls (**Figure 2—figure supplement 1A,D**). A Notch activity reporter, Notch Responsive Element (NRE)-green fluorescent protein (GFP) (**Stempfle et al., 2010**) was also upregulated in *Dl-/Dl-* clones as early as stage 2, and this expression persisted beyond stage 6 (**Figure 2C,D**), suggesting that NRE-GFP is probably more sensitive to Notch activation than Hnt in follicle cells. Together, these results suggest that Notch activity occurs independently of canonical ligands when both *cis*- and *trans*-ligands are removed, resulting in normal downstream developmental events in the follicle cells. Consistently, *Dl^RevF10^Ser^Rx82^* double mutant clones in the wing and eye discs show a slight cell-autonomous upregulation of NRE-GFP in the clone center, which would only occur if *cis*-inhibition blocked a DSL-independent mode of Notch activity, as interior cells have no access to *trans*-ligand (**Figure 2E,F**). This NRE-GFP

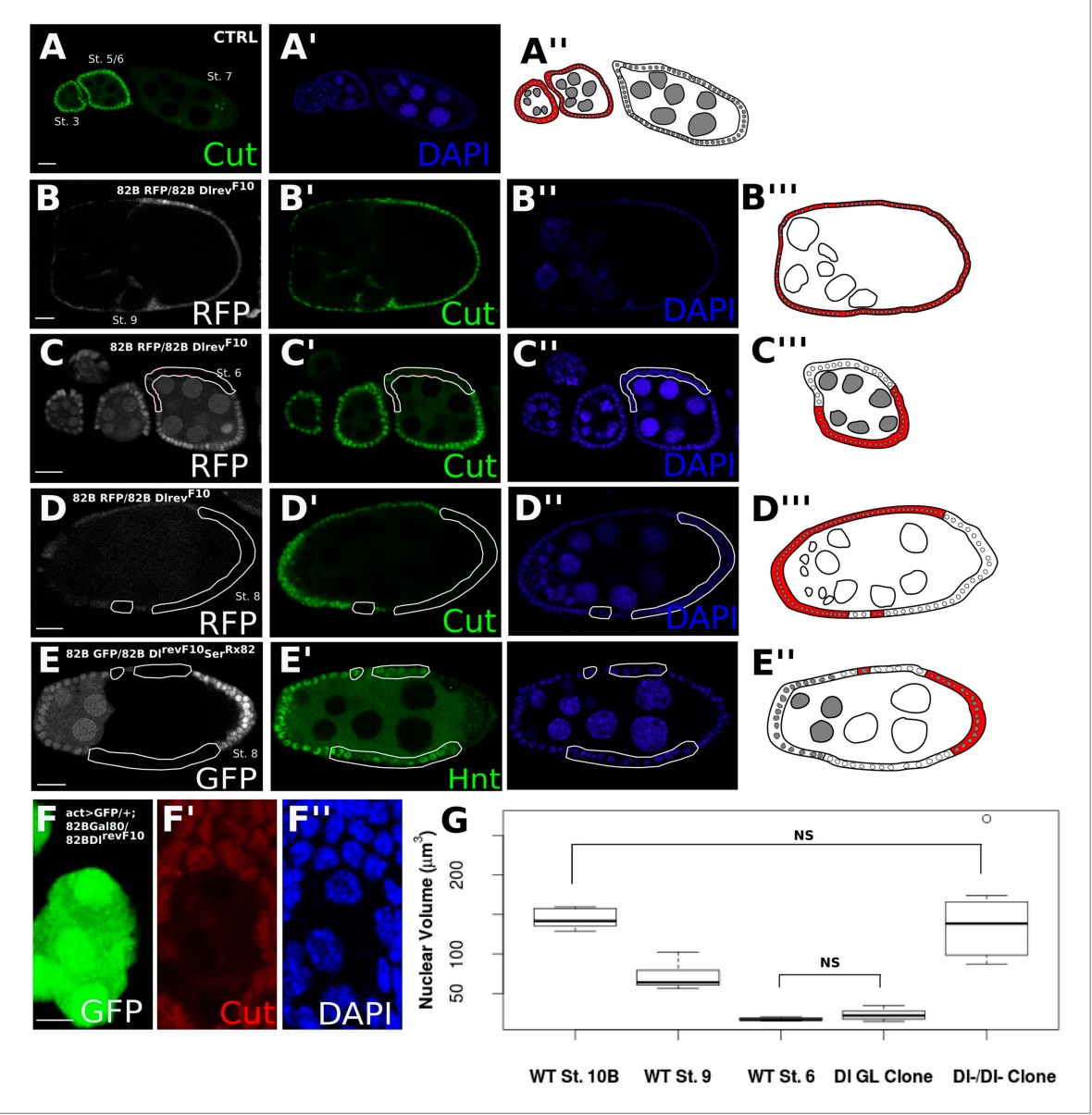

**Figure 1**. Follicle cells without DSL ligand bordering germline cells without DSL ligand show proper Notch activation and downstream differentiation. Illustrations legend: active Notch = white cytoplasm, inactive Notch = red cytoplasm, WT cell = grey nuclei, mutant clone = white nuclei. (**A**–**E**). Follicle cells downregulate *Cut* at stage 7 of oogenesis (**A**). *Dl^revF10* mutant germline cells cause late *Cut* expression in follicle cells (**B**). *Dl^revF10* mutant follicle cells downregulate *Cut* early (**C**). *Dl^revF10* follicle cell clones bordering *Dl^revF10* germline clones show proper *Cut* downregulation (**D**). *Dl^revF10Ser^Rx82* mutant follicle cell clones bordering *Dl^revF10Ser^Rx82* germline clones also show proper Hnt (**E**). See *Figure 1—figure supplement 2* for a z-series image for 1D and 1E. These germline/follicle cell clones (**D** and **E**) show increased nuclear size comparable to wild-type (WT) follicle cells which have entered the endocycle (n = 8 for each stage/genotype) (**F** and **G**). For (**G**), Welch t-tests were done to assess significance between each condition. The only comparisons that were not significant were between WT stage 10B and *Dl-/Dl-* clones and between WT stage 6 and *Dl* germline clones, indicating nuclear size in germline clones alone is similar to that of cells before the endocycle, whereas *Dl-/Dl-* clonal nuclei are more similar in size to cells that have entered the endocycle. Scale bars represent 20 µm, except in **F**, where the scale bar represents 5 µm.

The following figure supplements are available for figure 1:

**Figure supplement 1**. A schematic depiction of the early stages of Drosophila oogenesis.

**Figure supplement 2**. Z-stacked images of Dl-/Dl- clones and quantification of Cut staining in egg chamber clones.

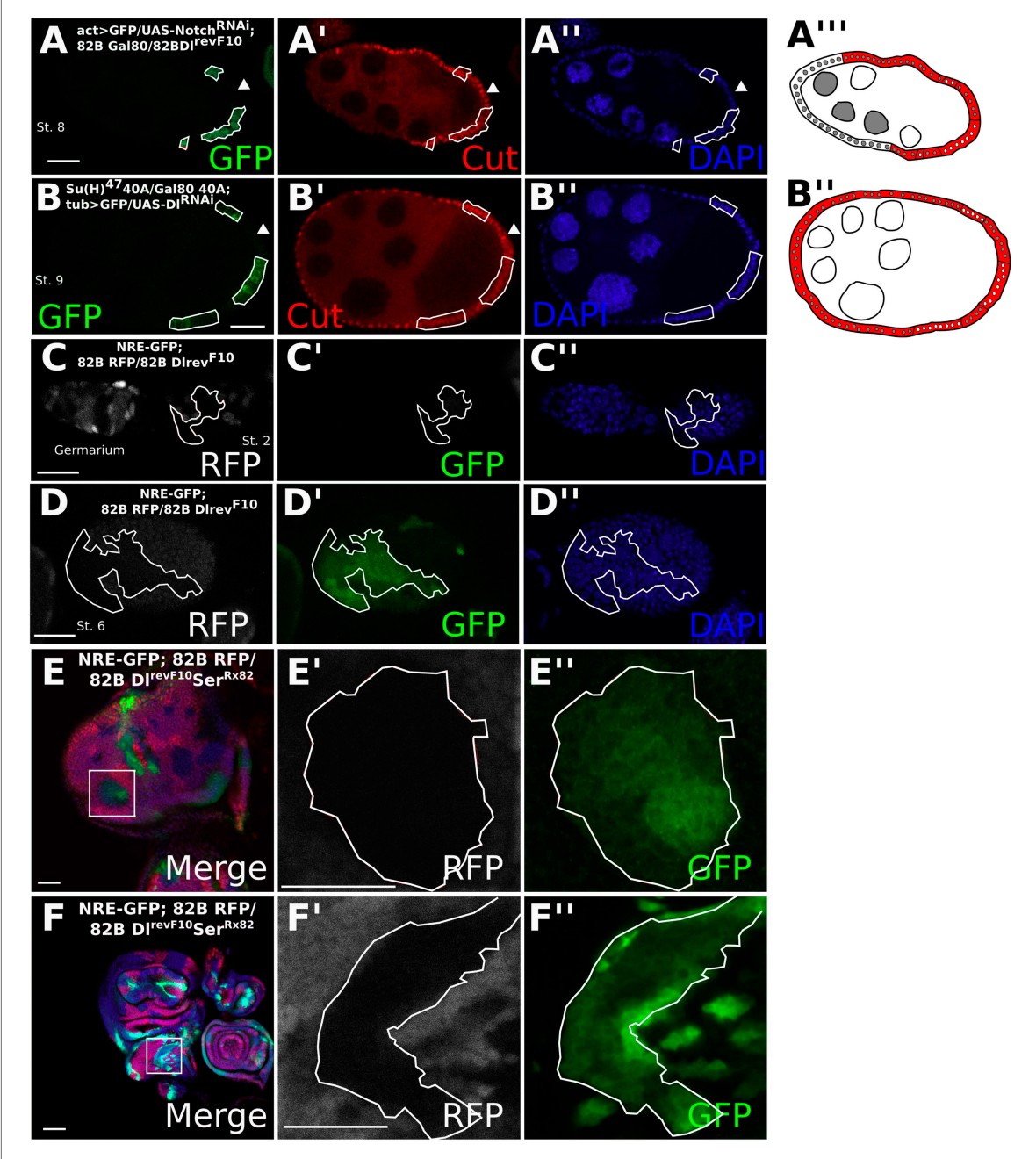

**Figure 2**. *Cis*-ligand represses ligand-independent Notch activity in the follicle cells and imaginal discs. *Dl*^revF10^ mutant MARCM germline/follicle cell clones co-expressing *Notch*^RNAi^ show prolonged *Cut* expression (**A**). *Su(H)*^47^ MARCM mutant germline/follicle cell clones co-expressing *Dl*^RNAi^ show failure to enter the endocycle (**B**). Germline clones are shown by late *Cut* expression in wild-type follicle cells (**A**, **B**, see arrowheads). See *Figure 2—figure supplement 1A* for control *Dl*^RNAi^-induced germline follicle cell clones. Notch Responsive Element-green fluorescent protein (NRE-GFP) is upregulated beginning from stage 2 (**C**) and through later stages (**D**) in *Dl*^RevF10^ germline and follicle cell clones. NRE-GFP is also upregulated cell-autonomously in *Dl*^RevF10^*Ser*^Rx82^ mutant clones in eye (**E**) and wing (**F**) imaginal discs. Scale bars represent 20 µm.

The following figure supplement is available for figure 2:

**Figure supplement 1**. Control experiments relating to *Figure 2*.

upregulation was spatially variable in the wing disc, having the highest prevalence in the notum region (25% incidence), a low incidence in the dorsal pouch (8%), whereas in the ventral pouch region it was never seen (n = 80) (*Supplementary file 1*), perhaps owing to the differential regulation of Notch degradation throughout the wing disc (*Hori et al., 2011*). As reported previously, most wing disc clones showed a higher NRE-GFP upregulation in the clone boundary where there is access to *trans*-ligand, indicating that the ligand-independent Notch activity observed occurs at a rather low level.

Drosophila S2 cells are reported to have no *Dl* expression and a very low level of *Ser* expression, which had no effect on Notch signaling (*Fehon et al., 1990*; *Graveley et al., 2011*) (*Figure 3—figure supplement 1*), and have been used as a model to study ligand-independent Notch activity (*Hori et al., 2011*). Upon transfection with *pMT-N^{FL}*, a $CuSO_4$-inducible full-length Notch construct, Notch activation was increased by a factor of 5.13 compared with the control cells, as indicated by a NRE-firefly luciferase reporter gene (p < 0.0001) (*Figure 3C*). Notch activation in S2 cells is at least partially dependent on endosomal trafficking, as double-stranded (ds) RNA against early endosome component, *Rab5*, or multivesicular body sorting protein, *hrs*, reduced the levels of Notch activation (*Figure 3A,B*). This is consistent with the in vivo studies indicating that ligand-independent Notch activation relies heavily on receptor trafficking (*Hori et al., 2012*) (*Rab5* p = 0.00623, *hrs* p = 0.0159), and our observation that Notch accumulates in *Dl-/Dl-* clones (*Figure 3—figure supplement 2*). A requirement for trafficking is consistent with the results of others who have demonstrated aberrant Notch activation in follicle cell mutants for trafficking components (*Wilkin et al., 2004*; *Vaccari et al., 2008*; *Schneider et al., 2013*), such as *tsg101* mutant clones, which show early Notch activation in the follicle cells (*Figure 3—figure supplement 3*). Furthermore, co-transfecting *pMT-N^{FL}* with *pMT-GAL4* and *pUASt-Ser^{del3}*, a form of *Ser* that cannot activate Notch, but only *cis*-inhibit, (*Fleming et al., 2013*) almost entirely abolished the Notch activation detected when N^{FL} was transfected alone (p = 0.0048) (*Figure 3C*). These results suggest that if Notch is expressed in a cell free of *cis*- and *trans*-ligands, DSL ligand-independent activity will occur and that *cis*-inhibition is extremely efficient in preventing this 'accidental' Notch activity as it travels through the endosomal pathway en route to degradation.

We next explored whether *cis*-inhibition can also block ligand-independent Notch activity induced in aberrant genetic backgrounds. The Notch target, *Wingless* (*Wg*) is normally expressed along the dorsoventral boundary of the wing disc (*Figure 4A*). *Lethal giant disc* (*lgd*) homozygous mutant (*lgd^{d7}*) larvae display overgrown imaginal discs and ubiquitous ligand-independent Notch activation in the wing pouch region, as shown by upregulation of *Wg* (*Figure 4B*). Notch activation in *lgd* mutant cells is caused by a defect in Notch trafficking and degradation, as the receptor is aberrantly transported to the limiting membrane of the lysosome which facilitates production of N^{ICD} (*Childress et al., 2006*; *Gallagher and Knoblich, 2006*; *Jaekel and Klein, 2006*; *Schneider et al., 2013*). Using dpp-GAL4 to misexpress *UAS-Dl* along the anterior–posterior axis of the wing disc in *lgd^{d7}* homozygous larvae, Wg expression was considerably reduced along the *dpp* expression domain, indicating that *cis*-inhibition can block the ligand-independent Notch activity observed in this situation (*Figure 4C*). Overexpression of *Deltex* (*Dx*), an E3 ubiquitin ligase that stimulates Notch monoubiquitination and promotes its trafficking to the lysosomal limiting membrane, has also been shown to induce ligand-independent Notch activation specifically in the ventral wing pouch region (*Matsuno et al., 2002*; *Hori et al., 2004*; *Wilkin et al., 2008*; *Schneider et al., 2013*) (*Figure 4D*). We used *patched* (*ptc*)-GAL4 to drive expression of *UAS-Dx* with either *UAS-Dl* or *UAS-Ser^{del3}*, whose ectopic expression leads to a reduction of Wg staining along the dorsoventral boundary (*Micchelli et al., 1997*; *Fleming et al., 2013*) (controls in *Figure 4—figure supplement 1A,E*). Co-expression of *Dx* and *Dl* led to a decrease in *Wg* expression in the ventral *ptc* domain as compared with expression of *Dx* alone (*Figure 4E*). When *UAS-Dx* and *UAS-Ser^{del3}* were co-expressed, there was a small but noticeable, albeit variable, decrease in Dx-induced Notch activation (*Figure 4—figure supplement 1B–D*). This incomplete reduction was probably due to the previously noted, slightly compromised, *cis*-inhibitory potential of *UAS-Ser^{del3}* (*Fleming et al., 2013*) (*Figure 4—figure supplement 1A*). Taken together, these results provide evidence that *cis*-ligand has a negative effect on the raised levels of DSL-ligand independent Notch activation incurred in genetically abnormal cells.

To quantify this effect, we co-transfected *pMT-Dx* with *pMT-N^{FL}*, causing an increase by a factor of 4.21 (p = 0.0021) in the Notch activation compared with transfecting *pMT-N^{FL}* alone (*Figure 4F*). Transfection of *pMT-N^{FL}*, *pMT-Dx*, *pMT-GAL4*, and *pUASt-Ser^{del3}* significantly (p = 0.0194) reduced the level of Notch activation (*Figure 4F*). We next treated cells with dsRNA for either *lgd* or *shrub* (a component of the ESCRT-III complex). *Lgd* dsRNA induced an increase in Notch activation by a factor of

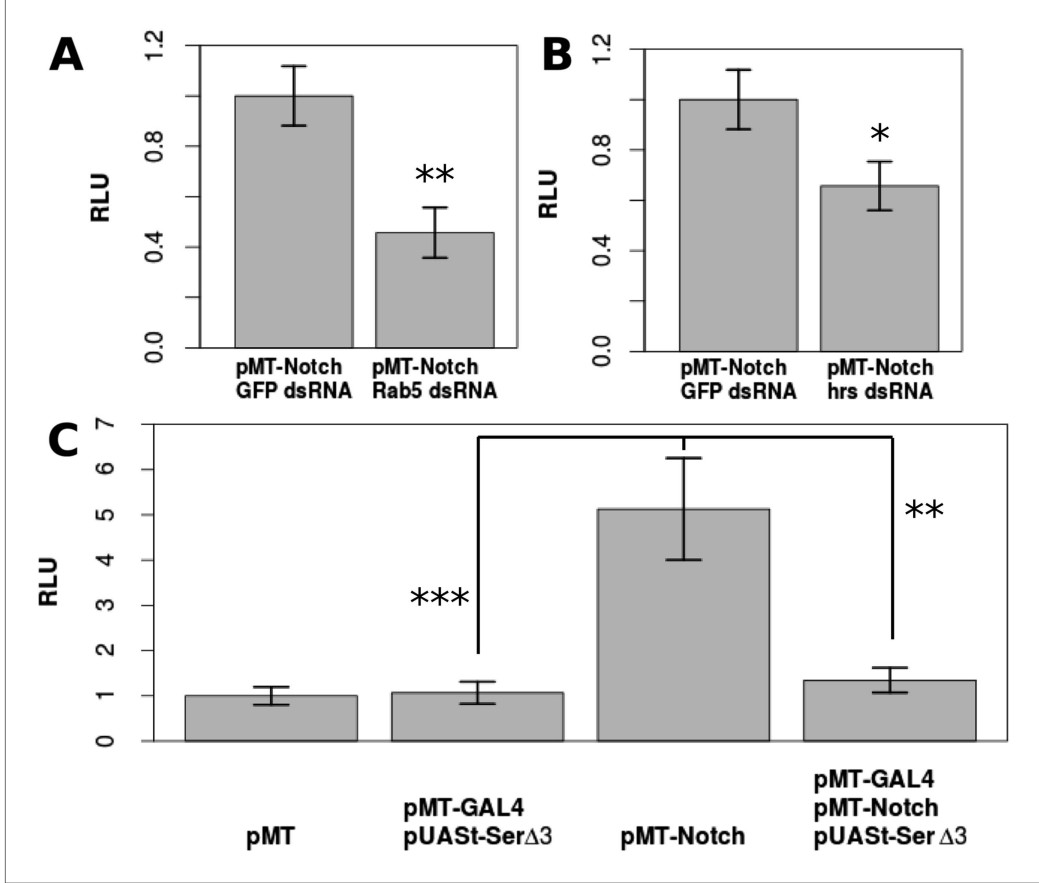

**Figure 3**. DSL-ligand-independent Notch activity in S2 cells is buffered by *cis*-ligand. Trafficking is important for Notch activation in S2 cells, as treatment with Rab5 dsRNA (**A**) or hrs dsRNA (**B**) significantly decreases the amount of Notch activated in S2 cells as shown by Notch-responsive luciferase activity (NRE-firefly) in relative light units (RLU). Transfecting only *pMT-Notch^FL* into S2 cells causes a 5.13-fold increase in Notch activation, which is almost entirely reduced (1.34-fold from the negative control) by co-transfection of *pMT-GAL4* and *pUASt-Ser^del3* (**C**). Each experiment was carried out with two technical replicates and three biological replicates. Means of the technical replicates were used to carry out a paired t-test (n = 3) for each comparison. Error bars represent standard deviation (SD).

The following figure supplements are available for figure 3:

**Figure supplement 1**. Addition of Ser dsRNA had no effect on the Notch activation in S2 cells in comparison with cells treated with control green fluorescent protein (GFP) dsRNA, indicating that the small amount of Ser expression is either not translated or does not significantly contribute to Notch activation upon transfection with *pMT-N^FL*.

**Figure supplement 2**. Notch accumulates in *Dl-/Dl-* clones.

**Figure supplement 3**. Follicle cells mutant for ESCRT component *tsg101* show early Notch activity in the follicle cells (***Vaccari et al., 2008***).

---

1.73 compared with GFP dsRNA-treated cells (p = 0.00286) (***Figure 4G***). Likewise, *shrub* dsRNA caused a 3.93-fold increase (p < 0.0001) in Notch activation in S2 cells (***Figure 4H***) (***Thompson et al., 2005***). Expression of *Ser^del3* in both situations led to a significant decrease in the amount of Notch activated in comparison with Notch-expressing cells treated with control dsRNA (*lgd* p = 0.0093, *shrub* p = 0.0257) (***Figure 4G,H***).

   To explore whether *cis*-acting ligands might block endogenous raised levels of ligand-independent Notch activation, in addition to the raised levels induced by genetic defects, we examined the effect of increased ligand expression in crystal cells in the larval lymph gland, which have recently been shown

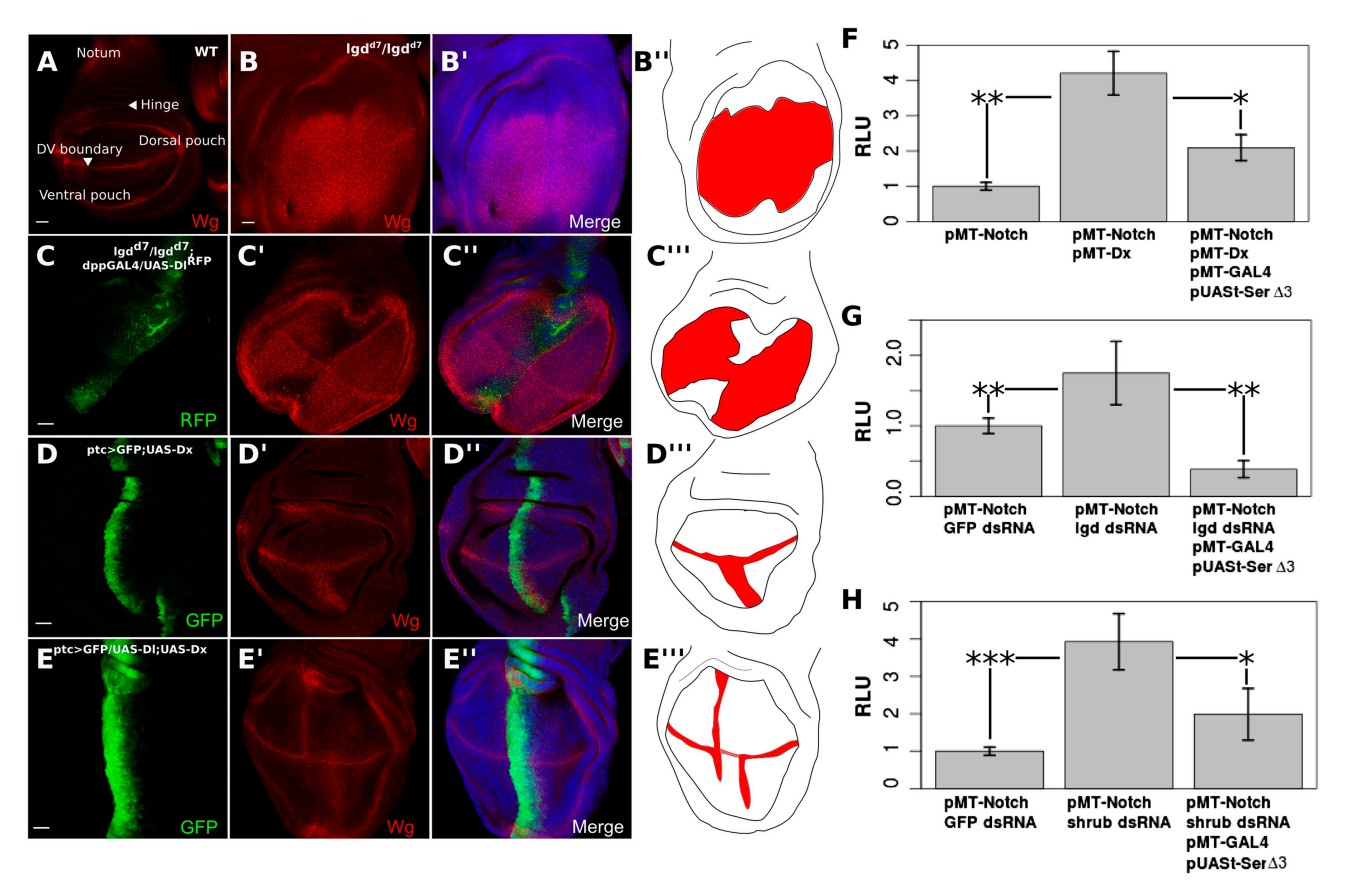

**Figure 4**. Notch ligand buffers against genetically induced DSL-independent activation. Wing discs were stained with Wg antibody and illustrations are colored red where Wg is expressed (**A**–**E**). A wing disc with regions of interest is labeled and WT Wg staining shown (**A**). *lgd^d7^/lgd^d7^* wing discs show ubiquitous *Wg* expression in the wing pouch as a result of DSL-ligand-independent Notch activity (**B**). Misexpression of *UAS-Dl* in *lgd^d7^/lgd^d7^* discs causes a reduction in Wg staining along the anteroposterior boundary of the pouch (**C**). *ptcGAL4* drives *UAS-Dx* causing ectopic Notch activity in the ventral wing pouch (**D**). Co-expression of *Dx* with *Dl* reduces Wg staining in the ptc domain (**E**), although, as in *lgd^d7^/lgd^d7^* discs, the reduction is not complete towards the dorsoventral boundary. *Cis*-ligand also decreases Notch activation caused by genetic defects in S2 cells (**F**–**H**). Co-transfection with *pMT-N^FL^* and *pMT-Dx* caused a significant increase in Notch luciferase reporter expression, and adding *Ser^del3^* significantly reduced this Dx-induced activation (**F**). Cells treated with lgd dsRNA (**G**) or ESCRT-III component, shrub, dsRNA (**H**) also caused significant increases in Notch reporter activity, either of which could be blocked by addition of *Ser^del3^*. For each of the S2 cell experiments, means were taken for technical duplicates and used for a paired t-test for three biological replicates. Error bars represent SD. Scale bars represent 20 μm.

The following figure supplements are available for figure 4:

**Figure supplement 1**. Co-expression of *UAS-Dx* and *UAS-Ser^del3^* has a variable effect on DSL-independent Notch activation.

**Figure supplement 2**. Endogenous DSL-independent Notch activity in crystal cells is reduced by *cis*-inhibition.

**Figure supplement 3**. Reduced Notch reporter activity in crystal cells was not caused by indirect effects on early ligand-dependent Notch signaling in prohaemocytes.

to have ligand-independent Notch activation (*Mukherjee et al., 2011*). Notch activity in crystal cells promotes cell survival, and decreased Notch activity leads to a 'bursting' phenotype (*Mukherjee et al., 2011*) (*Figure 4—figure supplement 2B,E*). Evidence for this bursting phenotype is provided by the disorganization of membrane-associated GFP (*Mukherjee et al., 2011*). Using *Lozenge* (*Lz*)-GAL4, a crystal cell lineage-specific driver (*Terriente-Felix et al., 2013*) to misexpress *UAS-Notch^RNAi^* or *UAS-Ser^del3^* led to a significantly higher proportion of cells showed the 'bursting' phenotype than wild-type crystal cells (*Notch^RNAi^* p = 0.0434, *Ser^del3^* p = 0.0286) (*Figure 4—figure supplement 2A,B,E*). Furthermore,

overexpression of *UAS-Ser^WT* led to a significant decrease of the Notch reporter *E(spl):mβ-CD2* expression in mature crystal cells (*Figure 4—figure supplement 2C,D,F*). Reduced Notch reporter activity was not caused by indirect effects on early ligand-dependent Notch signaling in prohaemocytes, as *Hnt*, a Notch target in differentiating crystal cells, (*Terriente-Felix et al., 2013*) was unaffected by ligand misexpression (*Figure 4—figure supplement 3A,B*). These observations indicate that increased ligand expression in crystal cells decreases cell survival by blocking Notch ligand-independent activation, and therefore the buffering role of *cis*-expressed ligand can be extended to endogenous cases of DSL-independent Notch activity.

In this study, we show that cells devoid of DSL ligands activate Notch sufficiently to stimulate reporter activity, and in the ovarian follicle cells the level of activation is above the threshold required to mediate normal Notch-induced downstream developmental events. During development, this type of noncanonical Notch activity is normally prevented by *cis*-expressed DSL ligands in numerous tissues. *Cis*-inhibition can also attenuate DSL-ligand independent Notch activity both in endogenous and genetically induced situations. Mechanistically, this could be explained if DSL ligands sequestered Notch at the membrane, made Notch more sensitive to degradation, or increased the stability of the heterodimer as it travels through the endosomal pathway. As we and others (*Fiuza et al., 2010*) have shown that increasing or decreasing ligand has variable effects on receptor distribution among tissues, and given that we observe a consistent effect among tissues on Notch activation upon *cis*-ligand removal, we prefer the stability hypothesis. *Fiuza et al. (2010)* show that ligand affects Notch stability during Notch activation by EDTA, giving support to the stability hypothesis as the most parsimonious explanation (*Fiuza et al., 2010*). It is suggested that retaining a pool of translated Notch receptor keeps the pathway in a condition capable of almost instant activation (*Sprinzak et al., 2010*). Therefore, we propose that a role of *cis*-ligands might be to keep the Notch pathway in a state of readiness by buffering against unintentional stochastic Notch activity resulting from normal processing through the endosomes. Endogenously, this may aid the ability of a cell to mediate future Notch-dependent developmental events that have strict temporal regulation.

# Materials and methods

## *Drosophila* stocks and generation of clones

The following fly stocks were used for Drosophila crosses. hs-flp[122];;FRT82B RFP (*Poulton et al., 2011*), FRT82B Dl[RevF10] (*Haenlin et al., 1990*), FRT82B Dl[RevF10]Ser[Rx82] (BDSC #6300), hs-FLP[122]; act-GAL4 UAS-GFP;FRT82B Gal80, UAS-Notch[RNAi] (VDRC #1112—no expression in germline cells), UAS-Delta[RNAi] (BDSC #34322—able to express in germline cells); hsFLP GFPstau; act > y[+] > GAL4, UAS-GFP, hs-flp[122]; Gal80 FRT40A; tubGAL4 UASGFP, Su(H)[47]FRT40A (*Morel and Schweisguth, 2000*), NRE-EGFP (BDSC #30727; *Stempfle et al., 2010*), ubx-FLP;;FRT82B RFP, patched-GAL4 UAS-GFP (*Hinz et al., 1994*), UAS-Dl[Myc] (a gift from Marc Muskavitch), tsg101[111019] from Kyoto stock center, UAS-Ser[WT] (BDSC #5815), UAS-Ser[del3−tom] (a gift from Robert J Fleming) (*Graveley et al., 2011*), UAS-Deltex (a gift from Martin Baron), lgd[d7]40A (BDSC #25087), dppGAL4 (BDSC #7007), lz-GAL4 UAS-GFP (BDSC #6314). To create FRT82B, Dl[RevF10] germline/follicle cell clones by the FLP/FRT or MARCM methods (*Golic and Lindquist, 1989*; *Lee and Luo, 2001*) (e.g., *Figures 1B,D–F, 2A,C–D, Figure 1—figure supplement 2A–B*), crossed flies were subjected to a 2 hr heat shock at 37°C for two consecutive days while in the mid-pupal to late-pupal stages. Flies were sorted three days after eclosion, and then kept for an extra three days at 25° before an additional 1-hr heat shock and incubation at 29°C with yeast paste for two more days before dissection. FLP-out-induced Dl[RNAi] germline/follicle cell clones (e.g., *Figure 2B, Figure 2—figure supplement 1A,B*) were produced by two consecutive 50-min heat shocks, followed by incubation at 25°C for a week and then transfer to yeasted vials in the 29°C incubator for dissection two days later. Evidence for MARCM and FLP-out-induced germline clones was provided by small nuclei and late Cut expression, as the UASt-GFP transgene does not reliably express in the germline. Follicle cell clones alone were produced by two 50-min heat shocks, followed by two days' incubation at 29°C (e.g., *Figure 1C* and *Figure 2—figure supplement 1C–D*). Imaginal disc FLP-FRT-induced mutant clones were produced either by a ubx-FLP or a 1-hr heat shock with hs-FLP[122] two days after egg laying. All other crosses were kept at 25°C unless otherwise noted. In lymph gland studies, Grubbs' test was used to identify significant (p < 0.05) outliers, which were omitted from further analyses.

## Immunostaining

Ovaries, imaginal discs, or lymph glands were dissected in phosphate-buffered saline (PBS), fixed in 10% formaldehyde, washed three times in PBS + Triton-X (PBT), and then blocked for at least 1 hr in

PBT with goat serum. Tissues were then either stained overnight with mouse anti-Cut (DSHB 2B10, 1:30), mouse anti-Hindsight (DSHB 1G9, 1:15), mouse anti-N$^{ICD}$ (DSHB C179C6, 1:15), mouse anti-N$^{ECD}$ (DSHB C4582H, 1:15), mouse anti-Wingless (DSHB 4D4, 1:20), mouse anti-Dl (DSHB C594.9B, 1:15), rabbit anti-βGal (MP Biomedical, Santa Ana, CA. SKU #08559761), or rabbit anti-GFP (abcam, Cambridge, UK. ab290—NRE-GFP was co-stained with this antibody to increase reporter sensitivity) primary antibodies. Tissues were mounted on slides after PBT washes and secondary antibody incubation. 4',6-Diamidino-2-phenylindole (DAPI) was used to stain nuclei. Samples were then analyzed with a Zeiss 510 or Leica SP2 confocal microscope and after analysis with the Image J software. Nuclear volume quantification was done with the Volumest plug-in for ImageJ.

## S2 cell transfection and RNA interference

S2 cells were grown under standard conditions and passaged once every three days in serum-free Gibco media (Invitrogen, Waltham, MA) supplemented with antibiotics. In preparation for transfection $10^6$ cells per milliliter were seeded into either 24-well plates or 96-well plates for experiments with or without dsRNA treatment, respectively. Transfections were carried out with Qiagen Effectene (Qiagen, Netherlands) transfection reagent according to the manufacturer's instruction. Plasmids used for transfection were pMT-Notch$^{FL}$ (a gift from Renjie Jiao), pMT-GAL4 (DGRC #1042), pUASt-Ser$^{del3}$ (a gift from Robert J Fleming), pMT-Deltex (a gift from Spyros Artavanis-Tsakonas), NRE-firefly luciferase (a gift from Sarah Bray), or Renilla luciferase (a gift from Sarah Bray). Aliquots (75 ng for 24-well plates or 50 ng for 96-well plates) of each non-luciferase plasmid were added and, where applicable, 10 ng of each luciferase plasmid. DNA concentration between transfections was kept constant with an empty vector. For experiments without dsRNA treatment, CuSO$_4$ was added to a concentration of 500 µM 24 hr after transfection, and cells were assayed 24 hr later. dsRNA was transcribed in vitro using the RiboMAX large-scale RNA production system-T7 kit (Promega, Madison, WI). The following primers were used to amplify genomic DNA taken from a single male fly from the NRE-GFP stock:

### Rab5
    Forward: GAATTAATACGACTCACTATAGGGCAGGGGACGAATTTCATTTG
    Reverse: GAATTAATACGACTCACTATAGGGAAAACCCTGCGCTTTCTTCT

### Hrs
    Forward: GAATTAATACGACTCACTATAGGGAATCGCCAACAATCAAGTCC
    Reverse: GAATTAATACGACTCACTATAGGGCGTGCAGCACTACTTTCCAA

### Lgd
    Forward: GAATTAATACGACTCACTATAGGGAGATGCCTCTGAGGAACCCGTCCAG
    Reverse: GAATTAATACGACTCACTATAGGGAGAGTGTGGGTTCTGGGGCAGCAGT

### Shrub
    Forward: GAATTAATACGACTCACTATAGGGACTTTTATGCAGGGACGTGG
    Reverse: GAATTAATACGACTCACTATAGGGTCCCTCGCTTCGAACTAAAA

### Serrate
    Forward: GAATTAATACGACTCACTATAGGGTCTCACCAACCAACCAATCA
    Reverse: GAATTAATACGACTCACTATAGGGCACAATATAGAGCGCGACGA

### GFP
    Forward: GAATTAATACGACTCACTATAGGGAGCTGGACGGCGACGTAAAC
    Reverse: GAATTAATACGACTCACTATAGGGATGGGGGTGTTCTGCTGGTAG

    Cells were treated with dsRNA at a concentration of 50 nM, and then transfected shortly after. CuSO$_4$ was added to a concentration of 500 µM later that day. Cells were incubatedfor five days, with an additional treatment of dsRNA on the fourth day.

## Luciferase assay

Cells were transfected with plasmids of interest together with an NRE-driving firefly luciferase expression and a constitutively activated Renilla luciferase to control for transfection efficiency. Luciferase measures were inspected with the Dual-Luciferase Assay Kit (Promega) in 96-well luminometer plates. Each transfection was performed in duplicate and repeated several times. Student's t test was used to test for statistical significance.

## Acknowledgements

Dongyu Jia discovered the phenotype of Hindsight, and Cut expression in Dl-/Dl- (Dl germline/follicle cell) clones, and developed the project as DSL ligand- independent mitotic cycle/endocycle switch initially. We thank Marc Muskavitch, Martin Baron, Robert Fleming, Renjie Jiao, Spyros Artavanis-Tsakonas, Sarah Bray, the Bloomington Drosophila Stock Center, the Vienna Drosophila RNAi Center, the TRiP at Harvard Medical School, and the Developmental Studies Hybridoma Bank for providing us with stocks and reagents. We also thank Yi-Chun Huang for technical assistance, and Gary Struhl, John Poulton, Pang-Kuo Lo, Gengqiang Xie, Jen Kennedy, Steven Lenhert, and Gabriel Calvin for helpful comments and suggestions while preparing the manuscript. W-MD is supported by NIH grants R01GM072562 and NSF IOS-1052333.

## Additional information

### Funding

| Funder | Grant reference number | Author |
|---|---|---|
| National Science Foundation | IOS-1052333 | Wu-Min Deng |
| National Institutes of Health | R01GM072562 | Wu-Min Deng |

The funders had no role in study design, data collection, and interpretation, or the decision to submit the work for publication.

### Author contributions

WHP, Conception and design, Acquisition of data, Analysis and interpretation of data, Drafting or revising the article; DJ, Conception and design, Acquisition of data; W-MD, Conception and design, Analysis and interpretation of data, Drafting or revising the article, Contributed unpublished essential data or reagents

## Additional files

### Supplementary file

• Supplementary file 1. Supplementary clonal data file. Excel worksheet containing clonal data from the egg chamber and the wing disc.

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
