## [Decision Letter]

Thank you for sending your work entitled “Cis-interactions between Notch and its ligands block ligand-independent Notch activity” for consideration at *eLife*. Your article has been favorably evaluated by Diethard Tautz (Senior editor), a Reviewing editor, and 3 reviewers.

The Reviewing editor and the reviewers discussed their comments before we reached this decision, and the Reviewing editor has assembled the following comments to help you prepare a revised submission.

In general, all reviewers agreed that this is an imaginative paper that addresses an important issue. We would like to see this paper published if the authors address the issues raised by the reviewers.

1) Repeatedly, in post review discussions, it came up that the paper was written in a confusing form in many places. To make this paper more understandable by the broad readership of *eLife*, it is essential that it is copyedited and simplified by the authors and perhaps in consultation with a non-*Drosophila* colleague. Also, many of the figures need to be improved.

2) In the Abstract, the authors state that the paper is about the ligand-independent N activity in the ovary; however, the suggested mechanism of DSL-independent N activation (endosomal-dependent N regulation) is shown only in the wing disk. If this mechanism is to be extrapolated to explain how ligand-independent N signaling acts in the ovarian soma, the authors should at least show that “endosomal pathway mutant” follicle cells also have precocious N activation.

3) Mosaic analyses in follicle cells provide evidence that these cells without both cis and trans ligand sources are still able to activate Notch through what is presumed to be a non-canonical pathway. It is not shown what the receptor does in these cells even though the consequences of its activation are well documented. I would have liked to see where the Notch receptor is under those conditions and presume that an immunocytological following of the Notch receptor could be informative. The conclusion that “Therefore, ligands expressed in the same cell as the Notch receptor have an endogenous role in buffering against DSL-independent Notch auto-activation” is too general and too strong and while this is possible it is certainly not conclusive. There are other roles that cis inhibition serves or indeed can serve.

---

## [Author Response]

*1) Repeatedly, in post review discussions, it came up that the paper was written in a confusing form in many places. To make this paper more understandable by the broad readership of eLife, it is essential that it is copyedited and simplified by the authors and perhaps in consultation with a non-Drosophila colleague. Also, many of the figures need to be improved*.

We have made substantial changes in the revision to clarify points that may have been a source of confusion before, especially for Figure 1 and the related writing, as it contains inherently complicated data (two cell types with a phenotype conditional on the clonal state of another). In order to increase the readability by non-*Drosophila* biologists, we have gone through the manuscript with non-specialists who have helped us to locate areas which may not be clear, and have attempted to clarify these points. For example, we added a figure to show the schematic drawing of oogenesis, which shows where the follicle cells and germline cells are located. We also include a WT wing disc stained with Wg for comparison and label regions of interest on this disc (i.e., Notum region, hinge region, DV boundary, wing pouch) which we refer to in the text.

*2) In the Abstract, the authors state that the paper is about the ligand-independent N activity in the ovary; however, the suggested mechanism of DSL-independent N activation (endosomal-dependent N regulation) is shown only in the wing disk. If this mechanism is to be extrapolated to explain how ligand-independent N signaling acts in the ovarian soma, the authors should at least show that “endosomal pathway mutant” follicle cells also have precocious N activation*.

Several earlier studies have reported that different trafficking mutants such as Su(Dx) (Wilkin et al, 2004), tsg101 (Vaccari et al, 2008), and lgd (Schneider et al, 2013) show precocious Notch activation in the follicle cells. We have added a sentence that makes these past results of others more explicit, and also added an example figure showing early Notch activity in *tsg101* mutant follicle cells.

*3) Mosaic analyses in follicle cells provide evidence that these cells without both cis and trans ligand sources are still able to activate Notch through what is presumed to be a non-canonical pathway. It is not shown what the receptor does in these cells even though the consequences of its activation are well documented. I would have liked to see where the Notch receptor is under those conditions and presume that an immunocytological following of the Notch receptor could be informative. The conclusion that “Therefore, ligands expressed in the same cell as the Notch receptor have an endogenous role in buffering against DSL-independent Notch auto-activation” is too general and too strong and while this is possible it is certainly not conclusive. There are other roles that cis inhibition serves or indeed can serve*.

We have included figures of Notch distribution in different stages of oogenesis in *Dl-/Dl-* clones, and noted that we did observe Notch accumulation, consistent with other cases of ligand-independent Notch activation. From the comments, we are not sure whether “an immunocytological following” also refer to a pulse-chase experiment (i.e. pulse with NECD antibody then chase for varying amounts of time before fixation and secondary antibody wash), this technique, although it works in the imaginal disc, has not been successful in the follicle cells, which was noted in Yan et al, 2009, *Dev Cell* 17. We agree with the reviewer that the conclusion we made on cis-inhibition might be too general and too strong. We therefore have reworded this to the following: “Therefore, we propose that a role of *cis*-ligands could be to keep the Notch pathway in a state of readiness by buffering against unintentional stochastic Notch activity resulting from normal processing through the endosomes.”